# Chromium Pollution in European Water, Sources, Health Risk, and Remediation Strategies: An Overview

**DOI:** 10.3390/ijerph17155438

**Published:** 2020-07-28

**Authors:** Marina Tumolo, Valeria Ancona, Domenico De Paola, Daniela Losacco, Claudia Campanale, Carmine Massarelli, Vito Felice Uricchio

**Affiliations:** 1Water Research, Institute-Italian National Research Council (IRSA-CNR), 70132 Bari, Italy; marina.tumolo@ba.irsa.cnr.it (M.T.); daniela.losacco@ba.irsa.cnr.it (D.L.); claudia.campanale@ba.irsa.cnr.it (C.C.); carmine.massarelli@ba.irsa.cnr.it (C.M.); vito.uricchio@ba.irsa.cnr.it (V.F.U.); 2Department of Biology, University of Bari, 70126 Bari, Italy; 3Institute of Biosciences and Bioresources, Italian National Research Council (IBBR-CNR), 70126 Bari, Italy; domenico.depaola@ibbr.cnr.it

**Keywords:** chromium, pollution, health risk, remediation

## Abstract

Chromium is a potentially toxic metal occurring in water and groundwater as a result of natural and anthropogenic sources. Microbial interaction with mafic and ultramafic rocks together with geogenic processes release Cr (VI) in natural environment by chromite oxidation. Moreover, Cr (VI) pollution is largely related to several Cr (VI) industrial applications in the field of energy production, manufacturing of metals and chemicals, and subsequent waste and wastewater management. Chromium discharge in European Union (EU) waters is subjected to nationwide recommendations, which vary depending on the type of industry and receiving water body. Once in water, chromium mainly occurs in two oxidation states Cr (III) and Cr (VI) and related ion forms depending on pH values, redox potential, and presence of natural reducing agents. Public concerns with chromium are primarily related to hexavalent compounds owing to their toxic effects on humans, animals, plants, and microorganisms. Risks for human health range from skin irritation to DNA damages and cancer development, depending on dose, exposure level, and duration. Remediation strategies commonly used for Cr (VI) removal include physico-chemical and biological methods. This work critically presents their advantages and disadvantages, suggesting a site-specific and accurate evaluation for choosing the best available recovering technology.

## 1. Introduction

Chromium is a transition metal that exhibits a complex chemistry. In water, chromium exists with oxidation states ranging from +6 to −2. The most stable forms are the hexavalent Cr (VI) and trivalent Cr (III) ones and can interconvert with each other [1]. Depending on the solution pH values, chromium can be encountered mainly as Cr (III) or Cr (VI) [2]. In nature, the oxidation of Cr (III) is not favoured because of the high E° value of the Cr (III)/Cr (VI) redox couple, only manganese oxide seems to be an effective oxidant in the environment. Otherwise, Cr (VI) can be easily reduced to Cr (III) by different reducing agents including Fe (II), phosphate, sulphide, and organic matter, for example, humic acid [3,4,5].

Public concerns with chromium are primarily related to hexavalent compounds owing to their toxic effects on humans, animals, plants, and microorganisms [6]. The risks for human health are dependent on dose, exposure level, and duration. A lasting and continuative exposure to chromium even at low concentration, that is, in the case of occupational exposure, can damage the skin, eyes, blood, respiratory, and immune system [7,8]. On a cellular level, the genotoxic effect of chromium leads to oxidative stress, DNA damages, and other harms that can result in tumour development [9,10].

Environmental contamination of Cr (VI) is gaining more consideration because it is widespread throughout the world with high levels in water and soil owing to natural and anthropogenic activities [11,12,13]. These include mining and metal works, steel and metal alloys production, paint manufacturing, wood and paper processing, dyeing, and raising the chromium content in wastewater [14,15,16]. In addition, the fall out of ashes produced by the incineration of coal or municipal waste for energy generation and the production of second-generation fertilizers contribute to the elevated Cr (VI) content in soil and water [17].

Chromium discharge limits in water are regulated on a national scale and often vary depending on the different type of industry or receiving water body (marine water, lake, river, sewer system). In Europe, the admissible concentration values of Cr (VI) as mg L^−1^ range from 0.05 to 2 according to the environmental policy of Norway and Poland (most precautionary value) and Netherlands [18,19].

Traditional approaches for Cr (VI) removal from water and wastewaters include physico-chemical methods such as chemical reduction, adsorption on porous surfaces with sites for ion exchange, and electrocoagulation. These strategies are also highly efficient for a high content of Cr (VI), but some limitations are related to sludge production, large amount of chemicals required, and consequent risk of secondary pollution [20]. To overcome these issues, bioremediation can represent a valid alternative, particularly with lower and moderate concentrations of Cr (VI) present in large volumes of wastewater [21,22,23]. Bioremediation techniques generally take advantage of the metabolisms of microbes, particularly bacteria, thanks to their high plasticity and widespread presence [24]. Bioremediation processes require bacterial resistance to pollutant and include intracellular and extracellular bioreduction, biosorption on cellular surface, and bioaccumulation. 

This review intends to provide an overview on chromium pollution in European Union (EU) waters at present, describing in detail its sources and health risks. This work also critically highlights the advantages and disadvantages of the main physico-chemical and biological techniques for Cr (VI) removal, developed in the last decade.

## 2. Chromium in the Environment: Natural Occurrence and Anthropogenic Source

Chromium exists in the environment in a number of valence states, whose most stable forms, Cr (VI) and Cr (III), are characterized by different properties [25]. The main source of Cr (III) in the environment is a natural one and it is related to chromite ore (FeCr_2_O_4_) [1,26]. This mineral present in mafic and ultramafic rocks is a complex of magnesium, iron, aluminium, and chromium in varying proportions, depending on the deposit [27,28,29,30,31].

Chromite in ultramafic rocks typically occurs as stratiform deposits, which may vary from less than a centimetre to 5–8 m. A secondary type of chromite deposit is known as podiform deposit, consisting of irregular pods or veinlets of aggregated chromite, often with nodular or orbicular textures [31]. Generally, chromite is chemically inert and insoluble in water, but microbial interventions together with other geochemical processes could promote the Cr (III) release in nature from chromite, increasing the possibility of its oxidation to Cr (VI) [26]. 

Chromite represents the main commercial form of chromium for industrial application. At industrial level the process involved in the Cr (VI) extraction from chromite has been known since the 19th century as oxidative roasting [26,28,30]. In the environment kinetic and several other factors, that is, pH and organic matter, mean that Cr (III) species dominate in nature. However, levels of Cr (VI) exceeding 70–90 µg L^−1^ in groundwater and water have been frequently measured as a result of man-made pollution [19,32,33]. Industrial use and urban source in fact are closely related to Cr (VI) accumulation in sediments and waters [34]. For example, the improper disposal of chromite ore processing residues, that is stocking them in open dump sites, results in a rapid migration by leaching of soluble contaminants into surface waters and groundwater [33,35]. The residue material is a mixture of finer particulate waste matter and fused material. The latter is finely porous with exposed chromite particles on the outer and inner surfaces, which tend to leach from all exposed surfaces [36]. Moreover, phosphate amendments produced from sewage sludge ashes or tannery sludge are rich in hexavalent chromium which can be leached. Their improper use is the main cause of chromium pollution in agricultural areas [37,38]. 

Although to a lesser extent than human activities, some geogenic processes also have an impact on the soil and groundwater Cr (VI) content [39], such as the weathering of ultramafic igneous and metamorphic rocks in several European countries (Greece, Italy, France, Serbia and Poland) [32,40]. For example groundwaters proximal to ultramafic rocks and sediments in La Spezia province, Italy, have a Cr (VI) content ranging from 5 to 73 µg L^−1^ exceeding the Italian limit for drinking water set at 5 µg L^−1^ as well as the World Health Organization limit for drinking water of 50 µg L^−1^ [32,41]. Birnessite is a Mn (IV) oxide-containing mineral, which commonly forms a coat onto weathered grains and fractures in Cr-rich ultramafic rocks. This mineral is associated with the Cr (VI) formation from natural Cr (III) in the environment [41,42]. Other Mn (IV) minerals involved in Cr (III) oxidation are asbolane, lithiophorite, hausmannite, and manganite. Particularly, in the last two minerals, the Mn (IV) reduction provides the most free energy for Cr (III) oxidation [43].

In the light of the above, hexavalent chromium concentration in groundwaters represents a combination of natural and anthropogenic factors, which are difficult to distinguish. They are summarized in Figure 1. 

According to the latest update of the European Pollutant Release register, a total of 512 facilities from EU countries are registered as they release chromium compounds into air and water [28,44]. These facilities mainly belong to the energy sector, including thermal power stations and mineral oil and gas refineries (on average, 27–80 mg per kg of oily sludge) [45]. Other industries are related to waste and wastewater management, metal production and processing (including metal ore, pig iron and steel), mineral and chemical production, paper and wood production and processing, tanning and dyeing [17,32,44,46,47,48,49]. The percentage contribution of the EU industries for Cr emissions in water is shown in Figure 2. Thermal power stations and other combustion installations contribute the most, followed by waste and wastewater management. Particularly, ashes generated as a waste material through combustion processes of coal, lignite, and municipal solid waste are rich in hexavalent chromium [39,50,51,52,53].

With regards to the tons of chromium and its compounds released in EU waters per industrial activity per year, see Table A1 of Appendix A.

## 3. Chromium Emissions and Discharge Limits for EU Member States

The European Directives 2000/60/EC and 2008/105/EC in the field of water policy and their subsequent amendments (Directive 2013/39/EU) did not identify chromium as a priority substance with regard to hazardous substances. Otherwise, chromium was listed in the Annex VIII of the Water Framework Directive as a main pollutant [54]. To date, no discharge limit has been established by the EU. Each Member State regulates chromium emission in the aquatic environment and national discharge limits often vary, according to the industrial type and the receiving water body [18]. Some governments regulate differently the concentration of total Cr from Cr (VI) supported by chemical, ecotoxicological, and epidemiological evidence [34,55,56,57]. In other countries, such as Greece, Austria, and Denmark, environmental policies regulate only the concentration of total chromium in water, according to studies that report a very high correlation between total chromium and the fraction of Cr (VI) [32,58]. The maximum discharge limits are 2 mg L^−1^ for Cr (VI) and 5 mg L^−1^ for total Cr in EU Member States, according to the policies of Netherlands and Spain and Belgium, respectively [18]. Particularly, the value of 2 mg L^−1^ for Cr (VI) in Netherlands refers to the discharge of wastewaters from paint and ink producing facilities. National discharge limits in water for most European countries are listed in Table 1. 

Data related to the Cr emission trend between 1990 and 2017 by EU countries, published by the European Environmental Agency (EEA), revealed a remarkable decreasing trend (−71%). On the other hand, a weak increase was measured in 2017 (+1.6%), with Germany, Poland, Italy, and the United Kingdom as major contributors for Cr emissions (23%, 11.4%, 10.8%, and 10% respectively, with regards to the total) [59]. It should be noted that, since February 2020, the United Kingdom is no longer an EU Member State.

## 4. Chromium Prevalent Forms in Aqueous Environment

Among the potentially toxic trace elements, chromium is the most common pollutant in groundwaters [60,61,62]. Chromium speciation in water depends on several factors, including organic matter, red-ox conditions and pH levels [55]. In general higher pH values favour the oxidation while lower pH values favour reduction [26,63]. 

The main free aqueous forms of Cr (III) are [Cr(OH)]^2+^, [Cr(OH)_2_]^+^, Cr(OH)_3_ (aq) and [Cr(OH)_4_] [1,3,34,64]. These ions can confer green colour to water. In natural groundwater pH ranging from 6 to 8 and [Cr(OH)_2_]^+^ prevails; under slightly acidic to alkaline conditions Cr (III) can quickly precipitate as amorphous chromium hydroxide, Cr(OH)_3_ (s) [1]. 

Compared to Cr (III), aqueous hexavalent chromium, Cr (VI), is the most oxidized, mobile, reactive, and toxic form with no sorption in most sediment at pH > 7. While acidity and other factors which increase the positive charge on soil colloids determine Cr (VI) adsorption and its removal from the liquid phase [64]. Adsorption of Cr (VI) usually decreases while pH increasing. Cr (VI) exists in solution as monomeric species H_2_CrO_4_, [HCrO_4_]^−^ (hydrogen chromate) and [CrO_4_]^2−^ (chromate) which gives a yellow colour to water when Cr (VI) concentration is greater than 1 µg L^−1^. The monovalent form predominates in acidic water while the divalent form predominates at neutral pH or above. In very acidic solution hexavalent chromium also exists as the dimeric ion [Cr_2_O_7_]^2−^ (dichromate) [1,65,66]. Normal environmental conditions favour the reduction of Cr (VI) to Cr (III) rather than the oxidation of Cr (III) to Cr (VI). Although Cr (VI) is thermodynamically stable only under oxidising conditions [67], the kinetics of reduction to Cr (III) under certain conditions can be slow [19]. Organic compound containing sulfhydryl groups and ferrous ions are common reductants, while the inorganic materials mostly involved into the natural oxidation of trivalent chromium in the hexavalent form are the manganese oxides [1,3]. Main forms of chromium in aquatic environments are listed in Table 2.

### Determination of Environmental Chromium

In recent years, techniques capable of determining total chromium and its oxidation states have been improved in order to optimize the detection limit and minimize the interconversion between Cr (III) and Cr (VI) which can occur during analytical procedures [55].

The analysis of total chromium means quantify the presence of Cr(III) and Cr(VI) in the dissolved and suspended fractions of a water sample. The analysis of total dissolved chromium is determined after filtration and preservation with nitric acid to a pH level below 2.0 to minimize precipitation [71]. Moreover, the determination of both dissolved and suspended fractions requires sample acidification to dissolve the suspended fractions. The acid digestion is necessary when the turbidity of the acid-preserved sample is higher than one nephelometric turbidity unit (NTU). After the digestion procedure, the sample is analyzed using several analytical methods such as atomic absorption spectroscopy (AAS), graphite furnace atomic absorption spectrometry (GFAAS) [72], inductively coupled plasma mass spectrometry (ICP-MS) [73,74,75], inductively coupled plasma atomic emission spectroscopy (ICP-AES) [76].

Cr (VI) can be determined in filtered solutions by colorimetric reaction with 1,5 diphenylcarbazide (DPC) at the wavelength of 540 nm [22,77]. Despite its simplicity, the method that uses DPC suffers from the presence of several interferents that can bring to an overestimation or underestimation of the values of Cr (VI), such as Fe (III), Fe (II), Hg, V, sulphides sulphates and organic matter present in the matrix [2]. In the EPA Method 218.7 ion chromatography followed by derivatization with 1,5-diphenylcarbazide and UV-VIS analysis is used for the detection of Cr (VI) in drinking water [78].

Other Methods using high-performance liquid chromatography (HPLC) coupled to ICP-MS have also been adopted for the quantification of both of Cr (III) and Cr (VI) in water samples [79]. 

The most used analytical techniques for chromium determination in water samples are listed in Table 3.

## 5. Health Risk 

Biological effects of chromium strongly depend on its oxidation state [56,57]. Cr (III) is a nutritionally essential trace element, nontoxic and poorly absorbed [87]. Mussels, Broccoli, wholemeal flour, garlic, basil, potatoes are just some recommended food for chromium intakes [62]. Chromium (III) picolinate is a common dietary supplement. Trivalent chromium enhances insulin activity functioning as receptor binding, and decreases the risk for diabetes mellitus [62,88]. As a consequence, its deficiency results in disorders in glucose metabolism and glucose intolerance [16]. Assuming a fractional absorption value of 25%, the daily requirement of absorbable Cr (III) is estimated to be 0.5–2 µg, provided by an intake of 2–8 µg per day of Cr (III) [89]. However, an excess quantity of trivalent chromium above the recommended value may result in a long-term toxicity and carcinogenicity [87].

Cr (VI) is the most toxic form [16], producing liver and kidney damage, internal haemorrhage and respiratory disorders. It has been characterized as carcinogenic to humans (Group I) by the International Agency for Research on Cancer [90]. Cr (VI) can enter the body when people breathe air, eat food, or drink water containing it. Fortunately, human body has compartments and mechanisms for attenuating Cr (VI) toxicity, related to specific reducing activities of body fluids [91]. These mechanisms involve the saliva, gastric juice, intestinal bacteria, blood, liver, epithelial lining fluid, pulmonary alveolar macrophages, peripheral lung parenchyma, and bronchial tree [16,91,92,93,94]. 

All the deleterious health effects of chromium observed in humans are dose, exposure level and duration dependant [62]. In 1984, Korallus et al. [95] demonstrated that human plasma could reduce spontaneously Cr (VI) ions of up to 2 mg L^−1^ to Cr (III) and this capability could be enhanced by assuming ascorbic acid. But the intake of Cr (VI) in excess of plasma reduction capability as well as of the red blood cells reducing capacity (at least 93–128 mg) determines hematological changes [94]. Likewise higher doses of Cr (VI) depress the phagocytic activity of alveolar macrophages and the humoral immune response, whereas lower doses of Cr (VI) stimulate phagocytic activity of the alveolar macrophages and increase the humoral immune response [96].

In addition, there is a great deal of the relative health effects of the various routes of exposure for Cr (VI) [16]. Occupational exposure to Cr (VI) by inhalation depends upon the job function and industry, but can reach several hundred micrograms per cubic meter and it is associated with lung cancer [1,62,90,97]. Chronic inhalation exposure to hexavalent chromium results in effects on the respiratory tract, damaging the nasal septum with perforations and ulcerations, causing bronchitis, decreasing in lung function, pneumonia, and nasal itching and soreness [98]. With regards to dermal absorption through skin exposure to hexavalent chromium, it may cause contact dermatitis, sensitivity, and ulceration of the skin [16,99]. Ingestion is the most significant source of exposure for polluted drinking water [100]. Even if Cr (VI) ingested can be reduced to its trivalent form by saliva and gastric juice, the most part remains as absorbable chromium. Thus, ingested Cr (VI) through contaminated food and water may produce effects on the liver, kidney, gastrointestinal, immune systems, and blood [94,96]. Ingesting less than 2 g of Cr (VI) compound can result in kidney and liver damage after 1–4 days of exposure, while a dose of 2–5 g of a soluble hexavalent chromium compound can be fatal to an adult human [1]. 

The dose received through ingestion of polluted groundwater of the Aosta Valley region, Italy, was calculated by Tiwari & De Maio [100] using Equation (1): ADD = (C_w_ × IR × EF × ED)/(BW × AT)(1)
where ADD represents the average daily dose, unit in mg/kg/day; Cw is the concentration of chromium in water, unit in µg L^−1^; IR is the ingestion rate, unit in L day^−1^; EF is the exposure frequency, unit in days year^−1^; ED is the exposure duration, unit in years; BW is the body weight, unit in kg; and AT is the averaging time (days) [100].

### Cellular Intake, Metabolism, and Toxicity of Cr (VI)

Cellular membranes are relatively impermeable to cationic trivalent chromium, as confirmed by in vitro and in vivo studies adding radioactive Cr (III) to whole blood [94]. 

On the contrary, Cr (VI) enters the cells by diffusion through a nonspecific anion channel [96]. Structural similarity of chromate ion to sulphate allows its easy entry through the general sulphate channels [42]. After crossing the cell membrane, chromium undergoes a series of metabolic reductions forming the unstable reaction intermediates, Cr (V) and Cr (IV), and finally the more stable form Cr (III) [94,101]. At physiological pH, intracellular reduction of Cr (VI) occurs involving several non-enzymatic and enzymatic antioxidants. Examples of non-enzymatic reducing agents are ascorbate (Asc), reduced glutathione (GSH) and cysteine (Cys) [102]. Asc reduces Cr (VI) via a two-electron reaction forming the reduction intermediate, Cr (IV). Reduction of Cr (VI) by GSH can be either by one- or two-electron reactions which produces Cr (V) or Cr (IV). Reduction by Cys is almost exclusively a one-electron reaction. A combined activity of Asc, GSH and Cys in cells reduces more than 95% of Cr (VI) into Cr (III) [101]. Other minor players in chromium intracellular reduction are cytochrome P450 reductase (only in absence of oxygen) and the mitochondrial electron transport complexes [9,62,101]. Thus, Cr (VI) is not directly responsible of genotoxicity. It does not react with macromolecules such as DNA, RNA, proteins and lipids [103]. The toxicity of hexavalent chromium within the cell is related to reduction process by generation of free radicals. Cr (VI), Cr (V), Cr (IV) and Cr (III) produce ROS through a Haber–Weiss reaction as shown in Figure 3 [4,9,104]. The intracellular oxidative stress produced by the above mentioned processes is directly or indirectly responsible of damages to macromolecules [42,96]. Cr (VI) metabolism products have been associated with the production of DNA-single strand or DNA-double strand breaks [45,105,106]. These can alter the function of cells leading to cancers. Also, the electrostatic interaction between stable Cr (III) species and negatively charged phosphate groups of DNA forms mutagenic and toxic Cr (III)-DNA complexes which affect the DNA replication and transcription [56]. The main pathways involved in chromium genotoxicity are summarized in Figure 4.

## 6. Remediation Strategies

The best-known disposal method of wastes and wastewaters enriched in chromium is often reduction of Cr (VI) to a less mobile and less toxic form, Cr (III), because Cr in industrial wastes occurs predominantly in the hexavalent form [27]. Some strategies used for chromium pollution remediation include chemical reduction methods by reducing agents such as Fe (0) and Fe (II), precipitation, adsorption, ion exchange, electrocoagulation, or biological reduction as a result of microbial metabolism [21,25,107,108,109].

### 6.1. Physico-Chemical Tratments

#### 6.1.1. Chemical Reduction

Many reducing agents have been applied (typically in acidic media) for the treatment of Cr (VI) polluted wastewaters and groundwaters. Reductants include ferrous compounds and zerovalent iron [3]. The experiment by Katsoyannis et al. (2020) [47] suggested an autocatalytic effect of Cr (VI) concentration on its reduction by ferrous iron. Moreover, multiple additions of Fe (II) in water spiked with Cr (VI) seem to be more efficient than adding all required Fe (II) at once. A previous study by Stylianou et al. (2018) [110] found as optimum a molar ratio Fe (II)/Cr (VI) of around 3 for chromium reduction, thus reducing the overall quantity of reductive reagents and produced sludge. Moreover, the zero-valent iron (ZVI) and particularly the nanoscale zero-valent iron (nZVI) are well documented in the scientific literature as a readily available and low-cost reducing agent for Cr (VI) removal [111], although the ecotoxicological effects on native microorganisms are rarely considered [112]. Another limit of this treatment is related to the possible aggregation of the nZVI particles, which lower its efficiency. To overcome the issue, in recent years, nZVI was applied using porous media, that is, bentonite and sepiolite as solid support [112,113]. Other reducing agents used in acidic media are reduced sulphur compounds, hydrogen peroxide, and hydrazine. Sodium hydrosulphite (dithionite) can be used directly in alkaline conditions, but is not usually cost-effective [3].

#### 6.1.2. Adsorption and Ion Exchange

Mechanical strength, osmotic stability and exchange capacity are characteristic of good adsorbents [114]. Parameters that influenced the adsorption rate are pH, adsorbent dosage, contact time and initial concentration of contaminant [115,116]. Choppala et al. (2018) [117] found that Cr (VI) sorption can be enhanced by adding inorganic amendments, such as elemental sulphur. The positive effect of elemental sulphur on Cr (VI) sorption is mediated through a pH decrease. The increase of Cr (VI) adsorption in acidic to slightly alkaline conditions can be explained as surface complexation reaction between Cr (VI) species and surface hydroxyl sites which are the sites for ion exchange [3,118]. 

Many researchers have focused on the application of natural and synthetic sorbents for chromium remediation including activated carbon, carbon nanotubes, modified clay and sand, and volcanic rocks such as pumice and scoria [6,21,36,119,120]. A class of well-known anionic clay minerals is layered double hydroxides (LDHs). These minerals consist of positively charged hydrotalcite-like host layers and charge-balanced interlayers of inorganic anions and water molecules that are apt to exchange with other anions owing to the weak interlayer bonding [115,121]. Moreover, biomaterials, such as raw materials derived from agriculture and forestry as well as biochar, can be applied as adsorbent materials due to their high porosity, surface area, and surface reactivity [12,122,123,124]. An important advantage of these kinds of sorbents is their inexpensiveness, which makes the process more sustainable. Another effective strategy for Cr (VI) removal includes the strong base anion exchange (SBA), which requires an inert polymeric resin activated with surface and interstitial exchangeable functional groups [125]. However, some considerations about efficient resin regeneration and waste minimization are important to improve operational, economic, and environmental performances of ion exchange [126].

#### 6.1.3. Electrocoagulation 

In the last decades, several studies [127,128,129,130,131,132] reported electrocoagulation processes for removing a high concentration of Cr (VI) from water and wastewater. Electrocoagulation is a process consisting of creating metallic hydroxide flocs inside the wastewater by electro dissolution of soluble anodes. Dermentzis et al. (2011) [133] found that some affecting parameters are pH, applied current density, and time, while initial Cr (VI) did not influence its removal rate by electrocoagulation. By contrast, Zewail et al. (2014) [130] as well as Genawi et al. (2020) [132] found that the initial chromium concentration determines the efficiency of the treatment. The electrocoagulation is quicker than the chemical coagulation and produces less slime and less dissolved salts. The most used pairs of electrodes are made by Fe–Fe, Al–Al, or Al–Fe [129]. In the case of iron anodes, the Fe (II) ions reduce Cr (VI) to Cr (III) in alkaline to neutral medium, while they are oxidized to Fe (III) ions, according to the following reaction: CrO_4_^2−^ + 3Fe^2+^ + 4H_2_O + 4OH^−^ → 3Fe(OH)_3_ + Cr(OH)_3_ [133]. Consequently, both the Fe (III) and Cr (III) combine with OH– ions, forming insoluble hydroxides which precipitate.

### 6.2. Bacterial Resistance and Remediation Capabilities

The wide metabolic diversity of microorganisms makes their application possible in reclaiming a number of contamination scenarios [134]. In particular, bacteria represent a highly promising and cost-effective resource for chromium removal owing to their high plasticity and widespread presence. They are able to reduce the toxic of chromium Cr (VI) to the less toxic trivalent state, both as a survival mechanism aimed at reducing toxicity around the cell and as a means of deriving metabolic energy for cell growth [22,135]. Other strategies useful for Cr (VI) removal, which take advantage of bacterial resistance to high pollutant concentrations, include bioaccumulation and biosorption [136,137]. It should be noted that resistance and reduction are found to be independent properties of bacteria. Not all Cr (VI)-resistant bacteria can reduce Cr (VI) to Cr (III). On the other hand, there are non-resistant bacteria that can reduce Cr (VI), although their growth is significantly inhibited at high chromate concentrations [23].

#### 6.2.1. Biosorption

It has been demonstrated that bacteria can facilitate the removal of metal species from aquatic solutions owing to adsorptive properties of their cellular surface [138,139,140]. The biosorption of heavy metal ions by microorganisms is influenced by several parameters including specific surface properties of the microorganism (biosorbent), the amount of biomass, physico-chemical parameters of the solution such as temperature, pH, initial metal ion concentration, and the existence of other ions [141]. Asri et al. (2017) [142] studied the biosorption potential of seven bacterial strains isolated from a polluted site, finding a high significant positive correlation between Cr (VI) removal by strains and their acceptor electron character γ^+^ (r = 0.90). Moreover, a significant negative correlation between the Cr (VI) removal potential and their donor electron character γ^−^ (r = −0.746) was observed. The presence of anionic ligands on bacterial cell wall (carboxyl, amine, hydroxyl, phosphate, and sulfhydryl groups) has a relevant role in metal sequestration from water [143].

Biosorption of heavy metals can be metabolically mediated (with ATP consumption) by living cells or a spontaneous physico-chemical pathway of uptake, which can occur both by living and dead cells [141]. Although the adsorption rate can be higher using dead biomass, living microorganisms are preferred for bioremediation because living cells are capable of a continuous metal uptake and self-replenishment [141,144].

#### 6.2.2. Bioaccumulation 

Cr (VI)-resistant/tolerant bacterial strains can also accumulate heavy metals within the cells and sequestrate them from the surrounding environment. The concentration of metals inside the cells can result from the interaction with surface ligands (biosorption) followed by passive or active transport into the cell [145]. The combination of active and passive uptake is called “bioaccumulation” [146]. In contrast to other metals, which occur predominantly as cationic species, chromium exists mainly in the oxyanion form (i.e., CrO_4_^2−^), and thus cannot be trapped by the anionic components of bacterial envelopes [147]. Owing to its similarity to SO_4_^2−^ anion, Cr (VI) can be easily transported across biological membranes via active sulphate transporters. Srinath et al. (2002) [148] found two bacterial strains isolated from tannery waste *Bacillus circulans* and *Bacillus megaterium*, able to bioaccumulate 34.5 and 32.0 mg Cr g^−1^ dry weight, respectively.

After its cellular intake, Cr (VI) undergoes reduction processes.

Raman et al. (2018) [149] studied the bioremediation potential of *Stenotrophomonas maltophilia* isolated from tannery effluent, revealing a Cr (VI) bioaccumulation rate higher than its reduction. 

#### 6.2.3. Bioreduction

Bacteria capable of reducing Cr (VI) mainly belong to nitrate-reducing, Fe (III)-reducing, and sulphate-reducing bacteria. Among the gram positive bacteria, *Bacillus*, *Deinococcus*, and *Arthrobacter* have shown Cr (VI) reduction capability [107,150,151]. Meanwhile, *Enterococcus*, *Shewanella*, *Pseudomonas*, *Escherichia*, *Thermus*, and *Ochrobactrum* are examples of gram-negative bacteria with potential application in bioremediation [152,153,154,155,156,157]. Microbial reduction is one of the most promising routes for in situ reclamation of Cr (VI) polluted groundwater [22,158]. 

Cr (VI) reduction is observed to occur both enzymatically and chemically via the reducing agents Fe (II) and H_2_S produced by bacteria [159]. Several compounds, such as cytochrome c on cell surface, intracellular nicotinamide adenine dinucleotide (NADH), extracellular polymeric substances such as extracellular protein, polysaccharide, and humic-like substances, may be involved in Cr (VI) reduction [160]. Particularly, cytochrome c is a heme protein localized in the inner and outer membranes, which is involved in the direct reduction of Cr (VI) to Cr (III) through electron transfer across the respiratory chain [161]. NADH is a coenzyme that functions as a hydride donor for chromate reductase to detoxify Cr (VI).

According to Ackerley et al. (2004), chromate reductase can be divided into two groups named class I and class II, based on sequence homology.

Two of the most studied class I reductase are ChrR and YieF [162,163]. ChrR from *Pseudomonas putida* is one of the best studied Cr (VI) reductases and has an NADH-dependent activity. A study on chromate stress in *Escherichia coli* by Ackerley et al. (2006) [164] using enzyme mutants revealed that ChrR protects against chromate toxicity, preventing chromate reduction by the cellular one-electron reducers, thereby minimizing reactive oxygen species (ROS) generation [165]. YieF is a dimeric flavoprotein that reduces Cr (VI) to Cr (III) through a four-electron transfer, where three electrons are consumed in Cr (VI) reduction and the fourth electron is transferred to oxygen [23].

The nitroreductases NfsA, a common enzyme in the genera *Bacillus*, possesses Cr (VI) reductase activity as a secondary function and belongs to class II [166]. This enzyme mediates one-electron reduction processes forming the Cr (V) intermediate, leading to high reactive oxygen species generation [152]. This secondary function is possibly the result of bacterial enzymatic adaptation to the relatively recent increase of Cr (VI) content in the environment caused by anthropogenic activities [165].

### 6.3. Comparison between Chemical and Biological Strategies for Cr (VI) Remediating

All the above-mentioned techniques (see Section 6.1 and Section 6.2) present advantages and disadvantages, as summarized in Table 4. Thus, the best choice for chromium remediation needs a site-specific and accurate evaluation [167]. This is true especially for in situ interventions, which require a thorough understanding of the geochemistry, hydrogeology, microbiology, and ecology of contaminated matrices [134]. Cost–benefit analyses are often used, occasionally in concert with comparative risk assessment, to choose between competing project alternatives [168]. The efficiency rate of each treatment can be affected by inherent properties of the polluted matrix and the achievement of the experimental conditions optimum. The efficiency of some physico-chemical and biological treatments reported in the recent literature is summarized in Table 5.

## 7. Conclusions

Chromium pollution of waters and groundwaters represents a serious environmental problem for EU countries. Thermal power stations and other combustion installations, followed by waste and wastewater management plants, are the most relevant industrial contributors to chromium emission in water. Among the possible forms of chromium, the hexavalent one is the most toxic because it can cause dangerous damage to human health. The genotoxicity of chromium, once it is introduced into human cells, can manifest favouring genomic instability, cancer onset, cell cycle arrest, and apoptosis. Consequently, the choice of the optimal remediation strategies for recovering waters and groundwaters from Cr pollution is a crucial step for both technicians and public administrators. In the last decade, several technologies have been tested in order to verify their efficiency in achieving chromium decontamination. Physico-chemical methods reveal high capabilities in removing this pollutant, but at the same time, show high costs of application. Differently, bioremediation approaches are more sustainable both in terms of costs than concerning the impacts on the treated matrices (waters/groundwaters), leading to no secondary pollution. Nevertheless, living organisms used for reclamation interventions can be inhibited by a higher concentration of the pollutant. Thus, there is no strategy that is the absolute best. A thorough understanding of the geochemistry, hydrogeology, microbiology, and ecology of the polluted matrix, together with a cost–benefit analysis, are required to choose between competing project alternatives. The comparison between physico-chemical and biological methods for Cr (VI) decontamination proposed in this work can be useful to evaluate and customize the opportune chromium remediation strategy.

## Figures and Tables

**Figure 1 ijerph-17-05438-f001:**
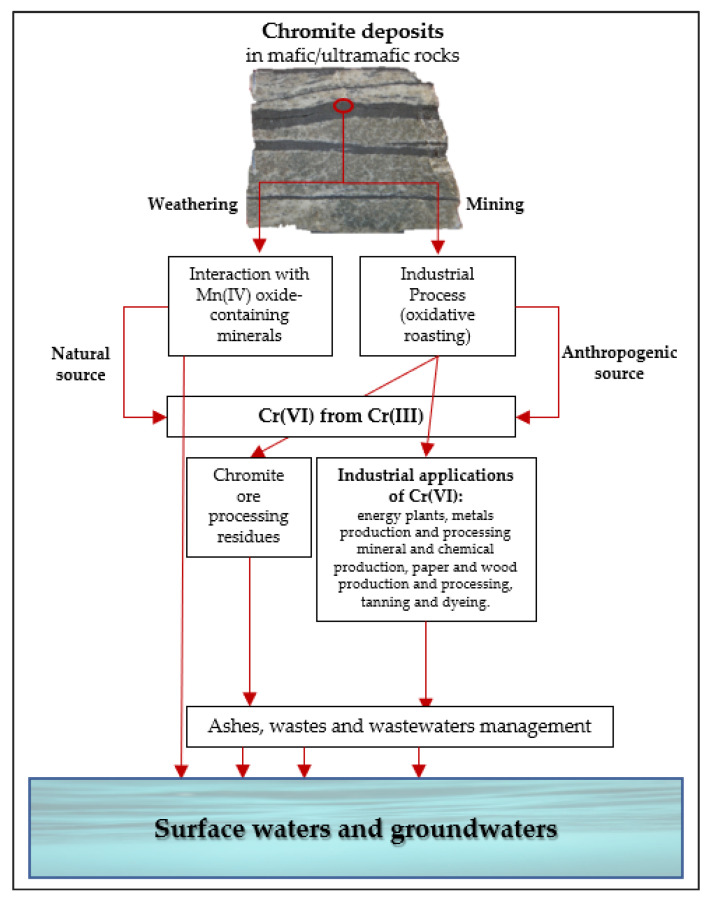
Schematic representation of the main sources of Cr (VI) in waters and groundwaters.

**Figure 2 ijerph-17-05438-f002:**
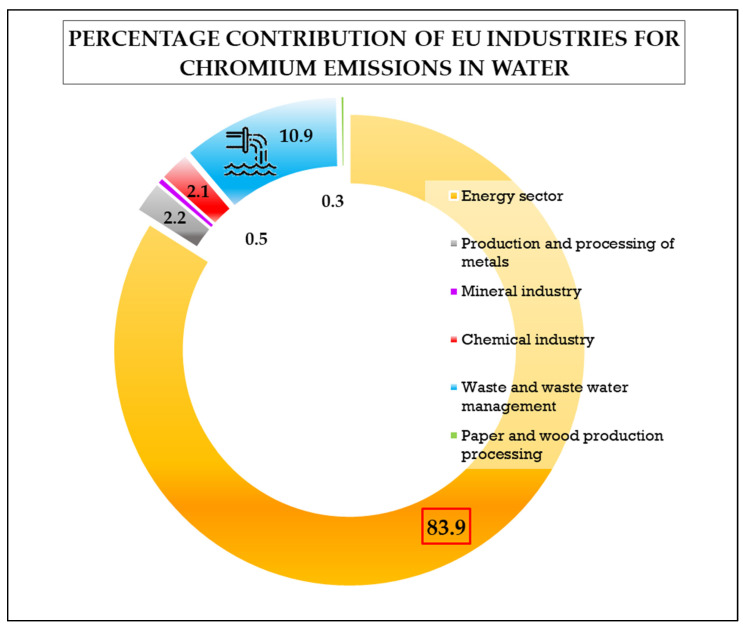
Percentage of Cr emission in EU waters per industrial sector. Data released by the European Environmental Agency are related to the year 2017 (data source: [44]).

**Figure 3 ijerph-17-05438-f003:**
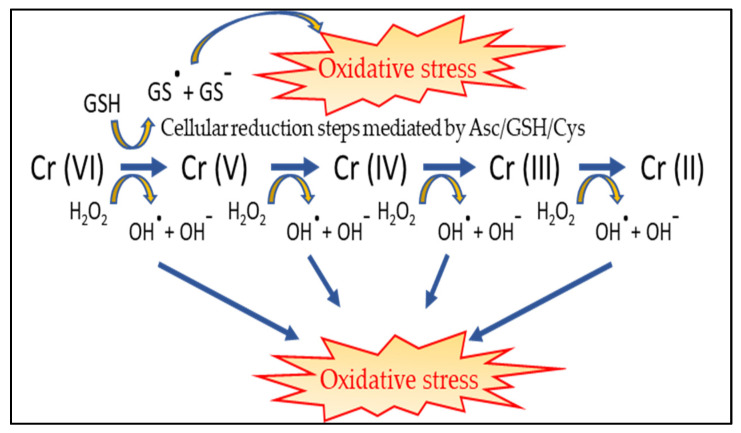
Schematic representation of free radicals formation during Cr (VI) reduction within the cell through Haber–Weiss reactions (modified from [15]). Asc, ascorbate; GSH, reduced glutathione; Cys, cysteine.

**Figure 4 ijerph-17-05438-f004:**
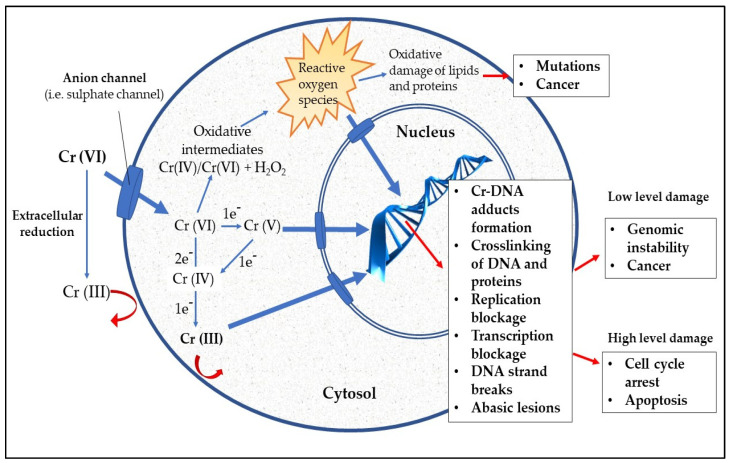
Cellular uptake and main pathways involved in chromium genotoxicity.

**Table 1 ijerph-17-05438-t001:** National discharge limits for total Cr and Cr (VI) concentrations in wastewaters, expressed as mg L^−1^ (adapted from [18]).

**Member State**	**Austria ^1^**	**Belgium ^1,2,3^**	**Croatia ^3^**	**Cyprus ^4^**	**Czech Republic ^1,2^**	**Denmark ^3^**
Total Cr	0.5–3	0.5–5	1–4	0.5	0.5–1	0.001–0.3
Cr (VI)	-	0.1–1	0.1	0.1	0.1–0.3	-
**Member State**	**Estonia ^4^**	**Finland ^4^**	**France**	**Germany ^1^**	**Greece ^3,^***	**Hungary ^1,3^**
Total Cr	0.5	0.7	0.5	0.1–0.5	0.6–1.5	0.2–1
Cr (VI)	0.1	0.2	0.1	0.05–0.5	-	0.1–0.5
**Member State**	**Ireland**	**Italy ^3^**	**Lithuania ^3^**	**Luxembourg**	**Malta**	**The Netherlands ^1^**
Total Cr	0.5	2–4	-	0.5	0.5	0.5
Cr (VI)	0.1	0.2	0.1–0.2	0.1	0.1	0.1–2
**Member State**	**Norway**	**Poland ^1^**	**Portugal**	**Sweden ^4^**	**Slovak Republic ^1^**	**Slovenia ^1^**
Total Cr	-	-	2	0.5	0.5–1	0.5–1
Cr (VI)	0.05	0.05–0.5	0.1	0.1	0.1	0.1
**Member State**	**Spain**	**Sweden ^4^**	**-**	**-**	**-**	**-**
Total Cr	5	0.5	-	-	-	-
Cr (VI)	0.3	0.1	-	-	-	-

^1^ Discharge limit varies depending on industrial type; ^2^ regional policy; ^3^ limit depends on the type of receiving water body; ^4^ case-specific regulation; * average monthly discharge limit (average daily discharge is double).

**Table 2 ijerph-17-05438-t002:** Chromium oxidation states and main forms in aquatic environments.

Oxidation State	Form	pH Condition	References
**Cr (III)**	Hexacoordinate complexes with complexing agents (i.e., water, ammonia, sulphate, urea, and organic acid)	0 < pH < 4	[2]
	Cr(H_2_O)_5_(OH)^2+^ abbreviated as [Cr(OH)]^2+^	slightly acidic conditions, 3.8 < pH < 6.3	[3,68,69]
	[Cr(H_2_O)_4_(OH)_2_]^+^ abbreviated as [Cr(OH)_2_]^+^	6 < pH < 8	[68,69]
	Cr(OH)_3_ (aq) *	slightly acidic to alkaline conditions	[69]
	Cr(OH)_3_ (s)	6.4 < pH < 11.5; max at pH ≈ 8	[1,66,69,70]
	[Cr(OH)_4_]^−^	pH > 11.5	[3]
**Cr (VI)**	H_2_CrO_4_	pH < 1	[66]
	[HCrO_4_]^−^	1 < pH < 6.4	[1,26,66,69]
	[CrO_4_]^2−^	pH ≥ 6.4	[1,26,66,69]
	[Cr_2_O_7_]^2−^	pH < 3	[66]

* sparingly soluble form, which tends to precipitate quickly.

**Table 3 ijerph-17-05438-t003:** Analytical methods for total Cr, Cr (III), and Cr (VI) in water samples. AAS, atomic absorption spectroscopy; GFAAS, graphite furnace atomic absorption spectrometry; ICP-MS; inductively coupled plasma mass spectrometry; ICP-AES, inductively coupled plasma atomic emission spectroscopy; HPLC, high-performance liquid chromatography; DPC, 1,5 diphenylcarbazide.

Sample Description and Cr Oxidation State	Procedure	Analytical Method	Detection Limit	References
Water, wastewater, and solid wastes (total dissolved Cr)	For the determination of dissolved Cr in a filtered aqueous sample aliquot, nitric acid is added to the sample, and then it is diluted to a predetermined volume and mixed before analysis.	ICP-OES	6.1 µg L^−1^	[80]
Groundwater, surface water and drinking water, wastewater, sludges, and soils (total dissolved Cr)	The same as the above procedure.	ICP-MS	0.08 µg L^−1^	[81]
Groundwater, surface water, drinking water, storm runoff, industrial and domestic wastewater (total dissolved Cr)	The same as the above procedure.	GFAA	0.1 µg L^−1^	[82]
Drinking water, groundwater, and water effluents (Cr (VI))	A filtered aqueous sample is adjusted to a pH of 9–9.5 with a buffer solution. A 50–250 μL aliquot of sample is introduced into ion chromatograph and separated on an anion exchange column. Post-column derivatization with DPC is followed by detection to 530 nm.	Ion chromatography associated with post-column derivatization and UV/VIS detection	0.3 µg L^−1^	[83]
Drinking water (dissolved Cr (VI))	Samples are analyzed by direct injection. An aliquot of 1 mL of sample is introduced into the ion chromatograph and Cr (VI) is separated from the other matrix components by an anion exchange column followed by derivatization with DPC.	Ion chromatography with post-column derivatization and UV/VIS detection	0.0044–0.015 µg L^−1^	[78]
Drinking water (Cr (VI))	A 2 mL aliquot of sample is transferred to a glass vial and sulphuric acid (1 mL 0.2 M) and DPC (1 mL 0.5% w/v) are added. Following, the absorbance is measured in microcuvettes with 1 mm light path at 543 nm against reagent blank.	Colorimetric method based on DPC dye for incorporation into a microfluidic detection system	0.023 µg L^−1^	[84]
Drinking water, surface water, and certain domestic and industrial effluents (dissolved Cr (VI))	Chelation of Cr (VI) with ammonium pyrrolidine dithiocarbamate (APDC) and extraction with methyl isobutyl ketone (MIBK) at pH 2.4. The extract is aspirated into the flame of the atomic absorption spectrophotometer.	AAS	2.3 µg L^−1^	[85]
Rain water, river water, spring water (Cr (VI))	Cr (VI) is collected as DPC complex on a column of chitin in the presence of dodecyl sulfate as counter-ion. The Cr-DPC complex retained on the chitin is eluted with a methanol–1 M acetic acid mixture, and the absorbance of the eluent is measured at 541 nm.	Preconcentration on a chitin column and spectrophotometric determination	0.05 µg L^−1^	[86]
Groundwater (Cr (VI)	A 25 mL aliquot of sample is added to 1 mL of 2.5 M H2SO4 and 1 mL of DPC 0.5%. The absorbance is measured after 10 min at 540 nm with a UV/VIS spectrophotometer using a cell with optical pathlengths of 10 cm.	Colorimetric assay using S-DPC	1 µg L^−1^	[22]
Drinking water (Cr (III) and Cr (VI))	On the basis of the type of ion exchange column used, HPLC is used to separate one of the two chromium forms. Following, a coupled ICP-MS is used to quantify the concentration of the species before and after the separation step.	HPLC-ICP-MS	0.005 to 0.5 µg L^−1^ (Cr (III)) 0.009 to 1.0 µg L^−1^ (Cr (VI))	[79]
Sea water (Cr (III) and Cr (VI))	A solid-phase extraction using anion exchange resins for Cr (VI) adsorption and chelating resins for Cr (III) adsorption is performed	ICP-MS	0.03 (Cr (III)) and 0.009 (Cr (VI))	[73]

**Table 4 ijerph-17-05438-t004:** Summary of the main advantages and disadvantages of common chemical and biological treatments for Cr (VI) removal. nZVI, nanoscale zero-valent iron.

Treatment	Advantages	Disadvantages	References
Chemical reduction with nanoscale zero-valent iron	High efficiency; high reactive surface area; easy to inject in aquifers.	Low stability; aggregation of nZVI particles; ecotoxicological effects on native organisms.	[111,112,113]
Adsorption coupled with ion exchange	Selective process; possible reuse of raw materials as green sorbents.	Complexity of adsorbents preparation; sludge generation; large amount of chemical required; waste generation; resin exhaustion; costly.	[126,131,169]
Electrocoagulation	High efficiency rate also with high chromium initial concentration; quicker and more sustainable than chemical coagulation processes; almost zero waste generation.	Skilled man-power requirement, several parameters influence its efficiency	[122,170,171]
Bioremediation	Cost-effective; ecological; sustainable; highly efficient with low and moderate pollutant concentration in large volume; no secondary pollution	Possibly inhibited by high pollutant concentrations;	[10,22,23]

**Table 5 ijerph-17-05438-t005:** Efficiency rate of some chemical and biological strategies reported in the recent literature.

Cr (VI) Initial Concentration	Treatment	Removal Efficiency (%)	References
50 µg L^−1^	Reduction by 1 mg L^−1^ of ferrous iron,	92%	[47]
300 µg L^−1^	Fe (II)/Cr (VI) in a molar ratio of around 3	Above 90%	[110]
0.6 mg L^−1^	Reduction by bentonite-supported nZVI	Above 90%	[112]
50 mg L^−1^	Electrocoagulation with Al-Al as pair of electrodes	42%	[128]
55.3 mg L^−1^	Electrocoagulation with Fe-Fe as pair of electrodes	91.7%	[131]
5 mg L^−1^	Electrocoagulation with Al alloy-Fe as pair of electrodes	98.2%	[130]
1 mg L^−1^	Adsorption onto modified carbon nanotubes	87%	[172]
30 mg L^−1^	Adsorption using biochar from Camellia oleifera seed shell	99.99%	[124]
0.5 mg L^−1^	Adsorption onto pumice (VPum) and scoria (VSco)	80% and 77%, respectively	[6]
50 mg L^−1^	Sulphur-based mixotrophic bio-reduction	95.5%	[160]
1000 µg L^−1^	Bioreduction by indigenous microorganisms enhanced by yeast extract addition	99.47%	[22]
50 mg L^−1^	Bioreduction by mixed bacterial consortium enhanced by phosphorus minerals addition	about 50%	[166]
100 mg·L^−1^	Biosorption using bacterial lawn deposited on membrane (seven bacterial strains tested)	from 5.32 to 99.87%	[142]

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
