# Peer review of "Chromium Pollution in European Water, Sources, Health Risk, and Remediation Strategies: An Overview"

_ijerph, 2020, doi:10.3390/ijerph17155438_

Round 1

Reviewer 1 Report

This manuscript presents a comprehensive review of the sources, health risk and remediation approaches pertinent to Chromium Pollution. In particular focussed on EU waters. The article is very relevant to the ongoing research and is imperative to the scientific community to have such manuscripts published to ensure there is a baseline for ongoing research.
The effects of Chromium pollution in water has been very well summarized in this manuscript. The manuscript is well organized and comprehensively discusses the pertinent information, background and also the data in sufficient detail.
However, some minor revisions must be made to ensure the manuscript is publishable and readable by the audience.
1) Figure 1 needs to be modified or a higher resolution image must be inluded. The scale bars are not legible at the current resolution.
2) There are minor typographical errors in the manuscript which needs to be fixed. eg: UE on line 122 needs to be fixed. Similar lines 149 and so on.
3) Figure 5 resolution should also be improved to ensure text in particular in the figure is visible and clear in the actual manuscript.

Reviewer 2 Report

In my opinion the present it is a good review about the Cr such as potential toxic metal occurring in water and groundwater as a result of natural and anthropogenic sources.

Before to accept the paper, I suggest to the authors to add a paragraph with the principal analytical methods used to analyze the Cr  

Reviewer 3 Report

Dear Authors,

The article raises many important issues related to the occurrence, properties, legislation, toxicity, and chromium removal.
Because the subject is pervasive, it is not easy to carry out a comprehensive literature review on several pages.
I suggest focusing on a narrower range of issues, e.g., only on health issues or on environmental impact (and removal of chromium compounds).

The article deals with a pervasive set of issues related to chromium compounds: starting from ownership and occurrence, through legislation, to toxicity and environmental aspects.

It is not possible to implement such a comprehensive topic on several pages of the literature review.
Therefore, I proposed a thorough change in the structure of the article by appreciating the authors' contribution to the preparation of the work.

Narrowing the subject to introduction (properties and occurrence of chromium compounds) and focusing only on health aspects or focusing only on environmental aspects. Thanks to this, it will be possible to carry out a more reliable literature review, not just to outline many topics.

I recommends re-referring the article for review after the authors after significant revisions.

Yours faithfully,
Reviewer

Round 2

Reviewer 3 Report

Dear Authors,

thank you for improving the manuscript.

Yours faithfully,

Rev